# Productivity losses and treatment cost of benign prostatic hyperplasia in Ghana

Daniel Senanu Dadee-Seshie[1], Sedem Godson Agbeteti[2], Banabas Kpankyaano[3], Aishah Fadila Adamu[1], Selase Kofi Adanu[4], Yakubu Alhassan[5], Maxwell Ayindenaba Dalaba[6], Kekeli Kodjo Adanu[7]*

1 Department of Surgery, Ho Teaching Hospital, Ho, Ghana, 2 Eastern Regional Hospital, Koforidua, Ghana, 3 Sunyani Teaching Hospital, Sunyani, Ghana, 4 Department of Environmental Science, Ho Technical University, Ho, Ghana, 5 Department of Biostatistics, School of Public Health, University of Ghana, Accra, Ghana, 6 Institute of Health Research, University of Health and Allied Sciences, Ho, Ghana, 7 Department of Surgery, School of Medicine, University of Health and Allied Sciences, Ho, Ghana

* kadan@uhas.edu.gh

## Abstract

Benign Prostatic Hyperplasia (BPH) is a common condition among ageing men, characterized by lower urinary tract symptoms that significantly impact the quality of life and economic productivity of affected individuals. The financial burden of BPH extends beyond direct medical expenses to include direct non-medical costs and productivity losses. In Ghana, limited data exist on the cost implications of BPH, leaving a critical gap in healthcare planning and resource allocation. We aimed to determine the cost and productivity losses associated with diagnosing and managing BPH at the Ho Teaching Hospital. A cross-sectional cost-of-illness study was conducted at the Urology Unit of the Ho Teaching Hospital. Data was collected from 105 patients diagnosed with BPH using structured questionnaires from 11th June 2024–30th September 2024. Direct monthly medical costs, including consultation, diagnostics, medication, and surgical interventions, were calculated using a bottom-up approach. Direct monthly non-medical costs covered transportation, food, lodging, and caregiver expenses, while productivity losses were estimated based on absenteeism and reduced working hours, using the human capital approach. The mean monthly direct medical cost per patient was GHS 890.11 (USD 60.35), with surgeries accounting for 30% of total expenses. The total direct non-medical costs, dominated by transportation (66.3%), amounted to GHS 9,842.00 (USD 667.25). Productivity losses due to absenteeism and caregiving responsibilities totalled GHS 4,746.28 (USD 321.78), with 30% of employed patients missing work. Notably, direct medical costs contributed the highest economic burden (86.5%), surpassing direct non-medical costs (9.1%) and productivity losses (4.4%). BPH imposes a significant financial burden on patients and households in Ghana, driven by high out-of-pocket medical costs, non-medical expenses, and lost productivity. The findings underscore the need for

**Data availability statement:** The dataset is available at figshare: https://doi.org/10.6084/m9.figshare.28528601.

**Funding:** The author(s) received no specific funding for this work.

**Competing interests:** The authors have declared that no competing interests exist.

cost-effective treatment strategies, improved health insurance coverage, and targeted interventions to alleviate financial hardships associated with BPH management.

## Introduction

Benign Prostatic Hyperplasia (BPH) is a common condition affecting elderly and middle-aged men, characterized by the enlargement of the prostate gland, which can lead to urinary symptoms [1]. The severity and combination of the symptoms can vary across individuals. Common symptoms include urinary frequency, nocturia, hesitancy, and poor urinary stream. BPH affects millions of men worldwide and presents a significant public health concern [2]. The global prevalence of BPH varies significantly across different regions and populations. Studies indicate that approximately 26.2% of men will experience BPH in their lifetime, with prevalence rates increasing with age [3]; and in Africa, prevalence rates ranging from 6.1% to 33.4% have been reported [4]. A study conducted in Ghana observed that 48.5% of participants aged 40 years and above had BPH, indicating a significant presence of the disease in the Ghanaian population [5]. Coupled with the high disease burden are the productivity losses and the cost implications of the disease.

One of the areas impacted is work productivity [6], which is lost because of absenteeism and presenteeism among employees with the illness. The associated lower urinary tract symptoms affect productivity in terms of the reduced hours spent at work compared to when they had no symptoms. Meanwhile, the cost of treating BPH has been on the ascendency. Studies conducted by Scott and Scott [7] in New Zealand marked an early effort to estimate the costs of treating BPH in both public and private sectors. By considering hospital data, the study reported the annual total direct medical costs for treated BPH in 1991 at 16 million New Zealand dollars. A more recent study estimated the annual direct medical cost of BPH to the UK economy at GBP 180.8 million [8]. Due to socio-economic factors, change in demography and innovations in surgery, the cost of treatment is likely to soar. An ageing population and rise in life expectancy are also expected to further worsen the cost of care.

In the US (United States), significant annual costs are reported for medications and diagnostic tests [9]. Long-term medications for BPH are more expensive than surgery [10]. Direct non-medical costs, such as informal care, transportation, and specialized products like absorbent pads, significantly contribute to the overall economic burden of BPH [11]. In Hungary, these costs account for 31% of the total treatment cost (EUR 876) [12]. Rural patients face high transportation costs [13], and caregiver expenses add further financial strain [14]. Dietary changes also impose costs, though specific figures are not well-documented [15].

Productivity loss (indirect cost) due to BPH significantly impacts the economy, costing the US an estimated 500 million out of the total annual cost of BPH USD 3.9 billion [8]. BPH interferes with daily activities [8,16] and patients experience low physical activity, absenteeism, and presenteeism, affecting work efficiency [12,17]. Despite its prevalence, comprehensive research on BPH-related productivity loss is sparse [18].

Literature on the treatment cost of BPH in Ghana is severely limited and to our knowledge, there is no comprehensive evidence on the subject. Yeboah [19] reported that the cost of BPH medications ranged from USD 300–500 while the cost of simple prostatectomy/Transurethral resection of the prostate (TURP) was estimated at USD 1,100. These estimates were, however, anecdotal at best, with no empirical data to buttress them. Due to this lacuna, relevant data to inform healthcare planning and resource allocation is lacking. In addition, expenditure on BPH is not only of public health concern because the household economy is equally implicated. Families make substantial out-of-pocket payments to defray the cost of care thereby forcing them to sacrifice other basic needs, sell assets and eventually become impoverished [20]. Thus, BPH treatment may have consequences on patients' quality of life and well-being, and to avert catastrophic health expenditure, interventions must be undergirded by empirical data.

To this end, this study determined the direct medical and non-medical costs and productivity losses of BPH at a tertiary health facility in Ghana, from the patients' perspective.

## Materials and methods

**Ethical statement:** Ethical approval was obtained from the UHAS Research Ethics Committee with certificate number: UHAS-REC A.9 [44] 23–24. A written informed consent was obtained from all study participants.

### Study design

This study used a cost-of-illness cross-sectional approach to collect data from the patients' perspective on the monthly cost and productivity losses of BPH at the Ho Teaching Hospital (HTH). A total of 105 participants were recruited from 11th June 2024–30th September 2024.

### Study setting

The study was conducted at the urology unit of the HTH. This unit runs an outpatient clinic that attends to an average of thirty (30) patients per day. Men diagnosed as having BPH and willing to participate were recruited.

### Sampling

To estimate the sample size, we relied on a previous study by Rencz et al [12], which looked at the cost of treating BPH in Hungary. We relied on this study due to the paucity of comprehensive data on the subject matter from Africa. This study was designed to consecutively select 99 patients, assuming a 95% confidence level and a 5% non-response rate, a standard deviation of 1829 and an error margin of 370 around the annual cost of BPH treatment.

Patients diagnosed of BPH, undergoing treatment at the urology unit of the HTH, who consented to participate were included in this study, while patients who had been diagnosed for less than 3 months were excluded. Patients with other co-existing conditions such as urethral stricture and bladder cancer were excluded. In addition, patients with confirmed prostate cancer were excluded in the analysis.

### Methods of data collection

This study used a quantitative data collection approach. The structured questionnaire comprised of three main sections: the sociodemographic characteristics of participants, the direct cost of treatment and productivity losses. The monthly direct medical cost was measured using a bottom-up approach, which involved quantifying each patient's medical expenses for BPH diagnosis and treatment in a month. This included costs for doctor visits, diagnostic exams, prescription drugs, surgeries, and other related healthcare services. The monthly direct non-medical cost comprises elements such as transportation costs, caregiver fees, and other related expenditures per month.

Productivity loss was estimated in terms of job loss due to BPH complications, absenteeism (days of work missed due to BPH-related symptoms), and time spent by family members or caregivers accompanying and caring for BPH patients during clinic visits.

Patients were consecutively recruited at the urology clinic after assuring them of confidentiality, voluntary withdrawal and data security. Questionnaires were administered through face-to-face interviews in privacy until the required number of patients was attained. Data was cleaned, validated and entered daily. Quality control measures instituted include pre-testing of questionnaires, training of research assistants and correction of errors before data entry.

## Data analysis

The data generated was entered into Excel 2021 and exported to SPSS version 27 for analysis. Descriptive statistical analysis was done using frequency distributions, percentages, means, median, ±standard deviations (SD) and presented in tables and charts. Regression analysis assessed the impact of the socio-demographic characteristics on cost elements, while Catastrophic Health Expenditure (CHE) was estimated using a 40% threshold as a primary measure and a 10% threshold to assess variations in impact. p-values less than 0.05 were considered statistically significant.

## Direct medical cost

The monthly direct medical cost included costs for doctor visits, diagnostic exams (for instance PSA testing, ultrasound), prescription drugs (e.g., alpha-blockers, 5 alpha-reductase inhibitors), surgeries (Transurethral resection of the prostate, open prostatectomy) and other related healthcare services. These expenses were summed up to determine the total direct medical cost and to obtain the average cost per patient, it was divided by the total number of respondents.

## Direct non-medical cost

The monthly direct non-medical cost was determined by summing up the cost of transportation, food and drinks, caregiver fees and fees for lodging/renting. The transportation cost was calculated by summing up the cost of travel by patient and caregiver to and from the hospital. Similarly, the cost of food and drinks was determined by the summation of expenses related to food and drinks while visiting the hospital. The fees for lodge/rent and caregiver fees were also tabulated. The total direct non-medical cost was estimated by adding up the various cost components. The average cost per patient was obtained by dividing the total cost by the number of respondents.

## Productivity losses

Productivity losses associated with the management of BPH was measured in terms of lost hours and absenteeism (days of work missed due to BPH-related symptoms), and time spent by family members or caregivers accompanying and caring for BPH patients in the hospital during clinic days at the expense of working hours. The number of workdays missed or productivity reduced was quantified and translated into monetary terms using the human capital approach; which estimates the value of lost productivity in terms of wages. Using the human capital approach, lost hours were multiplied by Ghana's minimum wage, which was GHS 18.15/day (USD 1.16/day) in 2024.

## Catastrophic health expenditure (CHE)

CHE due to treatment cost was also assessed. This cost element estimates the proportion of household income or expenditure spent on healthcare exceeding a certain threshold within a given period. Commonly used thresholds range from 10% to 40% [21–23]. In this analysis, a 40% threshold was applied as the primary measure, with a sensitivity analysis conducted using a lower threshold of 10% to assess the variation in impact. A patient was classified as having incurred catastrophic health expenditure if their treatment costs amounted to 40% or more of their monthly income.

In this study, the midpoint income range was used. This is because of the absence of data on specific individual incomes. The average overall cost per patient was estimated. The resulting value was then compared to the threshold income values to determine if CHE had occurred.

### Regression analysis

To assess the influence of socio-demographic factors on the cost of treating benign prostatic hyperplasia, an exploratory linear regression analysis was conducted. The analysis assessed the effect of the age, education, marital status, and income levels on the treatment cost.

## Results

### Socio-demography status

The descriptive statistics of the respondents are presented in Table 1. The average age of the participants was 71.0 years (± 8.48). The largest age group was 70–79 years (40.95%), followed by 60–69 years (29.52%), with a very small

**Table 1. Sociodemographic characteristics of study participants.**

| Category | Frequency | Percentage (%) |
|---|---|---|
| **Age (n = 105)** | | |
| 45-59 | 11 | 10.48 |
| 60-69 | 31 | 29.52 |
| 70-79 | 43 | 40.95 |
| 80-89 | 19 | 18.1 |
| 90+ | 1 | 0.95 |
| Mean age ± SD | 71.0 ± 8.48 | |
| **Marital Status (n = 105)** | | |
| Married | 75 | 71.43 |
| Widowed | 28 | 26.67 |
| Divorced | 2 | 1.9 |
| **Educational status (n = 105)** | | |
| Tertiary | 45 | 42.86 |
| SHS | 29 | 27.62 |
| JHS/Middle School | 15 | 14.29 |
| Primary | 10 | 9.52 |
| No Education | 6 | 5.71 |
| **Current Employment Status (n = 105)** | | |
| Unemployed | 65 | 63.81 |
| Self-Employed | 20 | 17.14 |
| Private Sector Employee | 12 | 11.43 |
| Public Sector Employee | 7 | 6.67 |
| Self employed | 1 | 0.95 |
| **Active NHIS (n = 105)** | | |
| Yes | 105 | 100 |
| No | 0 | 0 |
| **Physician Type (n = 105)** | | |
| Urologist | 85 | 81 |
| Primary Care Physician | 20 | 19 |

percentage (0.95%) being over 90 years old. The majority of participants were married (71.43%), while 26.67% were widowed, and 1.9% were divorced. Regarding education, 42.86% had tertiary education, and only 5.71% had no formal education. Employment status showed that 63.81% were retired, 17.14% were self-employed, and the rest were employed in either the private (11.43%) or public sectors (6.67%). Diagnosis was predominantly done by urology specialists (81%), with the remaining 19% diagnosed by primary care physicians, including house officers, medical officers, and surgical residents.

### Direct medical cost associated with the diagnosis and management of BPH

The mean direct medical cost was estimated at GHS 890.11(USD 60.35) per month, with diagnostic tests averaging GHS 278.80 (USD 18.90). The mean medications cost was GHS 200.74 (USD 13.61) and laboratory investigations (serum prostate specific antigen (PSA), full blood count, other relevant investigations specific to some patients) averaged GHS 79.14 (USD 5.37) per patient. Imaging studies (abdominopelvic ultrasound scan and other relevant investigations) costed GHS 52.29 (USD 3.55) and surgeries, GHS 264.76 (USD 17.95) per patients. No costs were reported for consultations since it is covered by the National Health Insurance Scheme (NHIS) and all respondents were enrolled onto the scheme. Overall, the total direct medical cost per participant averaged GHS 890.11 (USD 60.35), reflecting the diverse diagnostic approaches and medication needs among the study group. The cost estimations in USD are meant for international comparisons and due to wide standard deviations suggesting skewed data, the median cost was reported to allow for fair comparison of cost data. Further details are found in Table 2.

### Distribution of direct medical cost

Fig 1 provides a general overview of the various direct cost elements. Surgery accounted for 30% of the total direct cost build-up while diagnostic test and medications made up 29% and 22% respectively. 'Other' cost component includes the cost of catheter and gels which constituted 4% of the direct medical cost.

### Distribution of direct non-medical cost

Table 3 shows the direct non-medical costs associated with benign prostatic hyperplasia. They show the direct non-medical cost of BPH in categories: transportation, food and drinks, lodging/rent, home care services, and other non-medical costs. The total direct non-medical cost amounts to GHS 9, 842.00 (USD 667.25), with transportation being the highest individual cost at GHS 6,523.00 (USD 442.24), followed by food and drinks at GHS 3,029.00 (USD 205.36).

**Table 2. Direct medical cost.**

| Types of costs | Total GHS(USD) | Mean Cost GHS(USD) | Standard deviation | Median Cost GHS (USD) |
|---|---|---|---|---|
| **Direct medical** | | | | |
| Diagnostic tests (PSA+ TRUS+ Uroflow) | 27,044.00 **(1,833)** | 257.56 **(17.46)** | 145.04 **(9.83)** | 237.00 **(16.07)** |
| Medications | 21,078.00 **(1,429)** | 200.74 **(13.61)** | 225.30 **(15.27)** | 160.00 **(10.85)** |
| Lab Investigation | 8,310.00 **(563.34)** | 79.14 **(5.37)** | 71.03 **(4.82)** | 85.00 **(5.76)** |
| Imaging Studies | 5,490.00 **(372.23)** | 52.29 **(3.55)** | 67.13 **(4.55)** | 50.00 **(3.39)** |
| Surgeries | 27,800.00 **(1,884.76)** | 264.76 **(17.95)** | 1,883.26 **(127.68)** | 4,400.00 **(298.31)** |
| Other costs | 3,740.00 **(253.56)** | 35.62 **(2.42)** | 477.69 **(32.39)** | 50.00 **(3.39)** |
| Total direct medical cost | 93,462.00 **(6,336.41)** | 890.11 **(60.35)** | 2,869.45 **(194.54)** | 4,983.00 **(337.88)** |

*USD 1= GHS 14.75, Ghana interbank exchange rate, July 2024.

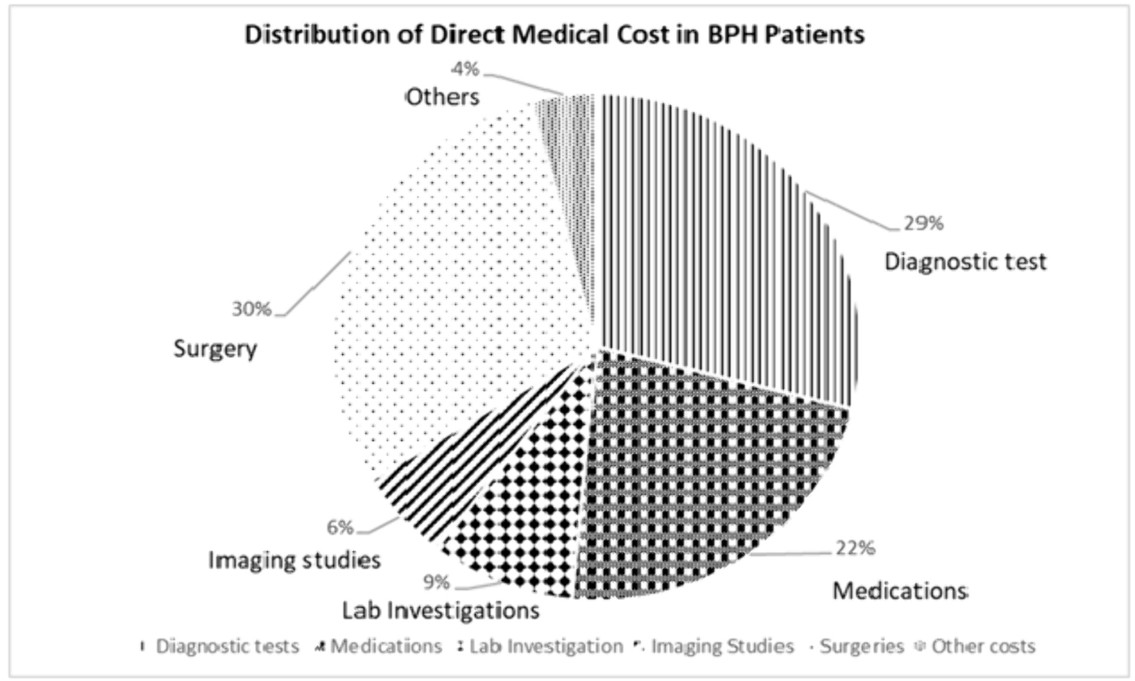

*Other cost in the figure below includes cost of catheter, gel and treatment from other sources including herbal and religious means.

**Fig 1. Distribution of direct medical cost.**

**Table 3. Direct non-medical cost.**

| Type of cost | Total GHS (USD) | Mean GHS(USD) | Std. Deviation GHS(USD) | Median GHS (USD) |
|---|---|---|---|---|
| **Direct non-medical cost** | | | | |
| Transportation per visit | 6,523.00 **(442.24)** | 62.12 **(4.21)** | 69.62 **(4.72)** | 40.00 **(2.71)** |
| Food and drinks | 3,029.00 **(205.36)** | 28.85 **(2.26)** | 85.00 **(5.76)** | 20.00 **(1.36)** |
| Lodging/rent | 200.00 **(13.56)** | 1.90 **(0.13)** | 0.00 | 200.00 **(13.56)** |
| Home care services | 90.00 **(6.10)** | 0.85 **(0.06)** | 21.21 **(1.44)** | 45.00 **(3.05)** |
| **Total direct non-medical cost** GHS(USD) | 9,842.00 **(667.25)** | 93.73 **(6.35)** | 175.83 **(11.92)** | 305.00 **(20.68)** |

*USD 1= GHS 14.75, Ghana interbank exchange rate, July 2024.

Lodging/rent and home care services have significantly lower costs at GHS 200.00 (USD 13.56) and GHS 90.00 (USD 6.10), respectively.

**Productivity loss associated with the diagnosis and management of BPH**

Table 4 shows the productivity losses associated with the diagnosis and management of BPH. It details various metrics, including days absent from work, hours spent travelling and waiting before consultation, reduction in working hours, and time spent by relatives on care and transportation. The total cost of productivity loss is GHS 4,746.28 (USD 321.78), with a mean of GHS 45.20 (USD 3.06) per patient. Notable figures include a total of GHS 2,069.10 (USD 19.71) lost per month

**Table 4. Productivity Losses Associated with BPH.**

| | Total hours | Total GHS (USD) | Mean GHS (USD) | Std. Deviation | Median GHS (USD) |
|---|---|---|---|---|---|
| Days absent from work | 912 | 2,069.10 **(140.28)** | 19.71 **(1.34)** | 132.63 **(8.99)** | 108.90 **(7.38)** |
| Hours Spent Travelling | 175.27 | 397.86 **(26.97)** | 3.79 **(0.26)** | 3.20 **(0.22)** | 2.28 **(0.15)** |
| Hours of waiting before consultation. | 321.41 | 729.60 **(49.46)** | 6.95 **(0.47)** | 3.20 **(0.22)** | 6.84 **(0.46)** |
| Reduction in working hours | 37.16 | 84.36 **(5.72)** | 0.80 **(0.05)** | 4.39 **(0.30)** | 0.00 |
| Days spent by relatives absent from work due to client illness | 296 | 671.92 **(45.55)** | 6.40 **(0.43)** | 8.93 **(0.61)** | 18.15 **(1.22)** |
| Hours spent by relatives on transportation | 134.59 | 305.52 **(20.71)** | 2.91 **(0.20)** | 3.91 **(0.27)** | 6.84 **(0.46)** |
| Hours household spend on care for clients | 214.94 | 487.92 **(33.08)** | 4.65 **(0.32)** | 8.80 **(0.59)** | 2.28 **(0.15)** |
| **Total** | **2,091.37** | 4,746.28 **(321.78)** | 45.20 **(3.06)** | 165.06 **(11.19)** | 145.29 **(9.85)** |

*USD 1= GHS 14.75, interbank exchange rate, July 2024.

due to absenteeism from work, and significant hours spent on activities such as waiting for consultation – 321.41 hours (3.061 hours on average per patient) and household care (214.94 hours).

The percentage distribution of costs associated with the treatment and management of BPH are as shown in Fig 2 above. Direct medical costs constitute the largest proportion at 86.5%, largely surpassing direct non-medical costs, which account for 9.1%. These two categories together represent the vast majority (95.6%) of the total costs, highlighting the financial burden borne directly by individuals or households. In contrast, productivity losses form a minimal share at 4.4%.

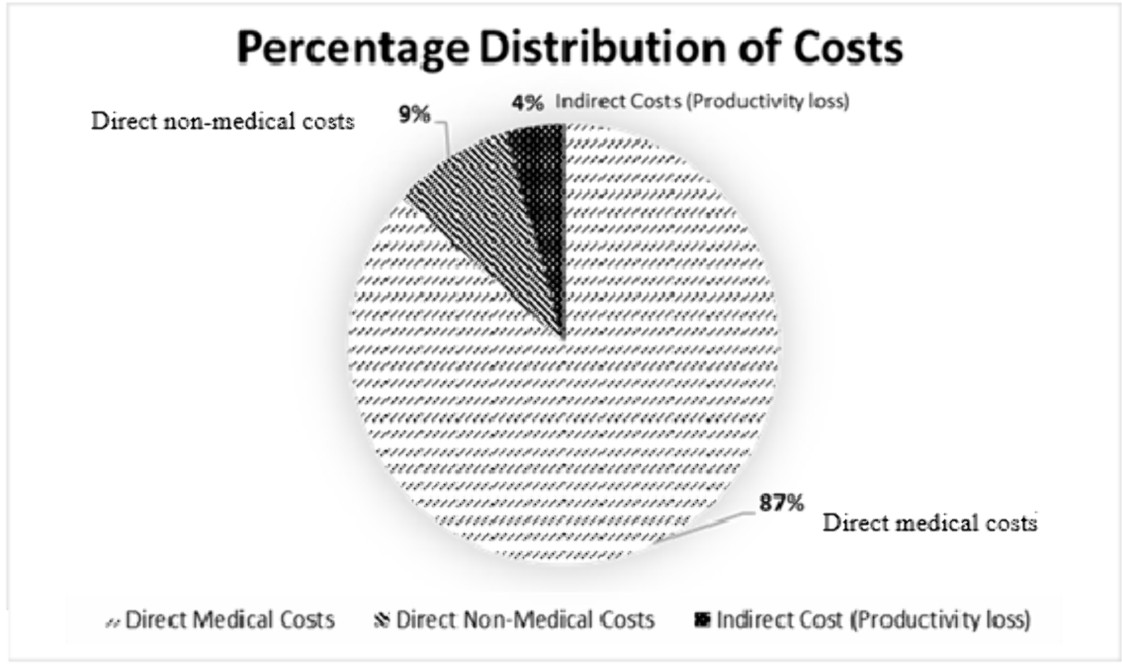

**Fig 2. Percentage composition of costs.**

## Overall cost (Direct cost and productivity losses)

Table 5 presents the overall cost of managing benign prostatic hyperplasia (BPH), comprising the direct medical, direct non-medical, and productivity losses expressed in both Ghanaian Cedis (GHS) and US Dollars (USD). The total cost incurred across all patients was GHS 108,050.28 (USD 7,325.44), with a mean cost per patient of GHS 1,029.05 (USD 69.78) per month. Direct medical costs accounted for the largest share of expenditure at 86.5% of the total cost. Direct non-medical costs contributed 9.11%, while productivity losses made up 4.39% of the overall expenditure. The mean monthly direct medical cost per patient was GHS 890.11 (USD 60.35), compared to GHS 93.73 (USD 6.35) and GHS 93.73 (USD 3.06) for direct non-medical and productivity losses, respectively.

## Catastrophic health expenditure

We evaluated CHE using the mean monthly cost per patient (USD 69.78) as a comparator.

Table 6 below presents the analysis of catastrophic health expenditure (CHE) among study participants (n = 105) across different income categories using both 40% and 10% thresholds.

Findings from the 40% threshold show that all individuals earning less than USD 136 per month (i.e., the two lowest income groups) experienced CHE, as the treatment cost exceeded 40% of their income. Conversely, individuals in higher income brackets did not experience CHE at the 40% threshold. When the sensitivity analysis was applied using a 10% threshold, nearly all income categories except the highest (USD 678/month) experienced CHE, reflecting a high level of financial burden on lower- and middle-income patients.

## Effects of socio-demographic variables on cost of treatment

Table 7 below presents the regression analysis of demographic factors associated with the total cost of BPH management. Age, marital status, educational status, and income level were included as predictors. The results show that none of these variables were statistically significant ($p < 0.05$). Age ($\beta = 41.52$, $p = 0.721$), marital status ($\beta = -62.18$, $p = 0.775$), and educational status ($\beta = -78.70$, $p = 0.365$) had weak associations with cost, while income level showed a relatively stronger

Table 5. Overall cost (direct cost and productivity losses of BPH management).

| | TOTAL (USD) | MEAN (USD) | STANDARD DEVIATION (USD) | MEDIAN (USD) | Percentage of total cost (%) |
|---|---|---|---|---|---|
| **Direct Medical Cost** | 93,462.00 (**6,336.41**) | 890.11 (**60.35**) | 2,869.45 (**194.54**) | 4,983.00 (**337.88**) | 86.50 |
| **Direct Non-medical Cost** | 9,842.00 (**667.25**) | 93.73 (**6.35**) | 175.83 (**11.92**) | 305.00 (**20.68**) | 9.11 |
| **Productivity losses** | 4,746.28 (**321.78**) | 45.20 (**3.06**) | 165.06 (**11.19**) | 145.29 (**9.85**) | 4.39 |
| **OVERALL COST** GHC (**USD**) | 108050.28 (**7325.44**) | 1,029.05 (**69.78**) | 3210.34 (**217.65**) | 5403.29 (**368.41**) | 100 |

Table 6. Catastrophic Health Expenditure.

| Income Category | Number of participants within this income category | Percentage (%) | Midpoint Income GHS (USD) | 40% Threshold GHS (USD) | 10% Threshold GHS (USD) | $69.78 ≥ 40%? (CHE?) | $69.78 ≥ 10%? (CHE?) |
|---|---|---|---|---|---|---|---|
| Less than $34 | 44 | 41.9 | 250.75 (**17**) | 100.30 (**6.8**) | 25.075 (**1.7**) | Yes | Yes |
| $34 – $135 | 41 | 39 | 1,246.38 (**84.5**) | 498.55 (**33.8**) | 1,24.638 (**8.45**) | Yes | Yes |
| $136– $338 | 11 | 10.5 | 3495.75 (**237**) | 1398.30 (**94.8**) | 3495.75 (**23.7**) | No | Yes |
| $339– $678 | 7 | 6.7 | 7500.38 (**508.5**) | 3000.152 (**203.4**) | 750.038 (**50.85**) | No | Yes |
| More than $678 | 2 | 1.9 | 11800.00 (**800- assumed**) | 4720.00 (**320.0**) | 1180.00 (**80.0**) | No | No |

**Table 7. Regression Analysis of Demographic Factors and Total Cost of BPH Management.**

| Variable | Coefficient (β) | Std. Error | t-value | p-value | 95% CI Lower | 95% CI Upper |
|---|---|---|---|---|---|---|
| Age | 41.52456 | 115.7478 | 0.36 | 0.721 | −188.1157 | 271.1648 |
| Marital status | −62.17693 | 217.0449 | −0.29 | 0.775 | −492.7878 | 368.434 |
| Educational status | −78.69582 | 86.52756 | −0.91 | 0.365 | −250.364 | 92.9724 |
| Income level | −153.1432 | 98.40206 | −1.56 | 0.123 | −348.3701 | 42.0837 |
| Constant | 1352.627 | 490.4736 | 2.76 | 0.007 | 379.5417 | 2325.713 |

negative relationship (β = −153.14, p = 0.123) but remained non-significant. This indicates a substantial baseline cost independent of demographic characteristics.

## Discussion

This study seeks to determine the monthly direct medical and non-medical costs and productivity losses of BPH at a tertiary health facility in Ghana. The results showed that the average monthly cost of treating BPH is USD 69.78. This is equivalent to an annual amount of USD 837.36 per patient.

The economic burden of diagnosing and treating BPH is considerable and varies based on healthcare setting. For instance, in Spain, annual costs for mild to severe symptoms ranged from EUR 124 to EUR 286 (average monthly cost of EUR10 – EUR 24), while in New Zealand, total direct medical cost for treatment was 16 million New Zealand dollars. More recent studies by Ahn et al [10], highlighted the significant financial implications of long-term BPH management, especially with continuous medical therapy compared to early surgical intervention. In this study, it was found that the total direct monthly medical cost was GHS 93,462 (USD 6,336.4) with a mean cost of GHS 890.11 (USD 60.35) per patient. Diagnostic tests, including PSA, Transrectal ultrasound (TRUS), and uroflowmetry, were associated with substantial costs, totalling GHS 27,044 (USD 1,725.54) with a mean cost of GHS 278.80 (USD 17.79) per patient. Medication costs amounted to GHS 21,078 (USD 1,345.67), with a mean of GHS 200.74 (USD 13.61), accounting for 22% of the total direct medical cost, which is consistent with other studies indicating medication as a major financial burden [10]. Although medications are a major contributor to overall costs due to their long-term nature, surgeries have a higher per-patient cost. This is typified by the relatively high surgery cost in this study, accounting for 30% of the total direct medical cost output. Compared to findings from other studies conducted by Kovacs et al [8], the costs of treatment in this study are lower, reflecting differences in healthcare pricing and availability of diagnostic services. However, the total annual direct medical cost (USD 76,036.92) is much higher than findings in neighboring Nigeria, where the cost was estimated at USD 12,800 [24]. Nonetheless, the financial burden on patients in Ghana is significant, especially given the lower average income levels and the out-of-pocket nature of healthcare expenditure.

The direct non-medical cost of BPH summed up to GHS 9,842.00 (USD 628.19), with an average cost of GHS 93.73 (USD 6.35) per patient. Transportation was the largest expense at GHS 6,523.00 (USD 416.24), averaging GHS 62.12 (USD 4.21) per patient, with variability attributed to wide differences in travel distance to and from the hospital facility, and means of transport preferences. The median cost of transportation, which is GHS 40.00 (USD 2.55) highlights that while many patients experience low transportation cost, others faced significant burdens. Food and drinks were the second largest expense, amounting to GHS 3,029.00 (USD 193.19) with an average cost of GHS 33.29 (USD 2.12) per visit. Lodging costs were negligible, with only one patient reporting an expense of GHS 200.00 (USD 12.76) as most patients were able to see a doctor on the same day of visit to the hospital. Usage of home care services was minimal as only two patients utilized the service at a total cost of GHS 90.00 (USD 5.74) per month. These findings align with studies, by Rencz et al [12] and Zou et al [13], where transportation was highlighted as a major cost for BPH patients. Although costs in the present study are lower compared to studies conducted elsewhere, they are significant in the local economic context since the

minimum wage in Ghana is significantly low (USD 1.16); and taking into consideration that approximately 81% of participants in this study had experienced catastrophic health expenditure. This emphasizes the need to address transportation and food expenses as critical components of the financial burden.

This study also highlights the significant economic and social burden of productivity losses associated with BPH treatment. Among 40 employed participants, 30% missed work and 27.5% reduced their working hours due to BPH. These figures are indicative of the pervasive impact BPH has on individuals' ability to maintain steady employment. Global literature consistently underscores the disruptive nature of BPH, with studies by Garraway et al [16] revealing that BPH significantly interferes with daily activities, which often translates into lost productivity at work. In the context of Ghana, where formal employment is not always accessible and many individuals rely on daily wages or informal work, the impact of BPH can be even more pronounced. The total work hours lost amounted to 912 hours, translating to GHS 2,069.10 (USD 132.04) per month, with an average loss of GHS 137.94 (USD 8.80) per patient. The burden extended to caregivers, who experienced an estimated GHS 971.92 (USD 62.02) in productivity losses, with an average of GHS 24.88 (USD 1.59) per month, alongside additional caregiving and transportation costs. This study underscores the cascading effects of BPH on households, and overall quality of life in Ghana, where financial losses can have severe implications due to low average incomes, as exemplified by mean hourly earnings of USD 1.01 in Ghana [25]. A related study among a rural community in Uganda found that most men are unable to seek medical care due to the high cost associated with treatment. The few men who sought care did so only because of the severity of symptoms and its interference with their daily lives. These findings emphasize the need for targeted interventions to reduce the economic and social burden of BPH, particularly in resource-constrained settings [26].

The direct medical costs constitute a substantial 86.5% of the total financial burden, consistent with findings from Rencz et al [12] and Ahn et al [10], which emphasize the significant impact of medication and physician visits on overall healthcare expenses. Notably, Ahn et al [10] reported that medication costs alone often exceed 50% of total treatment expenses in long-term medical therapy. In a similar study, Igwe & Eshiet [24] estimated the direct annual cost of managing BPH in Nigeria at USD 12,800 with the main cost drivers being medications and laboratory investigations. In contrast to our study, which found direct non-medical costs to account for only 9.1% of the economic burden of BPH patients, Rencz et al [12] reported a higher proportion of 31%. This variation may be attributed to differences in transportation costs, accessibility of healthcare facilities, reliance on home care services, and lodging requirements. Additionally, productivity losses in our analysis were relatively low, contributing only 4.4%, compared to the 23% reported by Rencz et al [12], where productivity losses represented a more significant share of the financial burden. This discrepancy may stem from variations in healthcare systems, employment structures, and workplace policies governing individuals affected by BPH.

An exploratory regression analysis from this study showed that sociodemographic factors (age, marital status, educational status, and income) did not predict the total cost of BPH management indicating that other factors influenced the total cost of treatment. This aligns with findings by Rencz et al [27], who reported that while men with BPH incurred higher costs than those without the condition, demographic variables (such as age) did not strongly or consistently explain cost differences. A separate study from Ghana found that costs incurred are high relative to patients' incomes, but did not demonstrate strong predictive relationships of demographic factors like education or marital status on cost variation [28]. Another cost-of-illness study in Hungary likewise revealed that disease severity (measured by symptom scores) was one of the strongest predictors of cost, rather than sociodemographic traits [29]. Disease severity was not analyzed in this data. Other treatment-related variables, such as transportation and medication costs, which may be regarded as significant cost drivers were also not included in this analysis. Be that as it may, findings from this study are in alignment with a broader literature suggesting that while demographic factors may show trends, they often do not reach statistical significance unless accompanied by clinical or system-level variables.

This study found that catastrophic health expenditure (CHE) is heavily concentrated among the lowest-income groups (all participants earning less than USD 136/month faced costs exceeding 40% of their income; constituting 81% of the

study population) and nearly all income categories (except the highest) exceeded 10% thresholds, underscoring high financial vulnerability among lower- and middle-income patients. These findings are consistent with a recent systematic review and meta-analysis for sub-Saharan Africa which found a non-negligible pooled incidence of CHE and stressed that poorer households and those with chronic or service-intensive needs are at far higher risk [20]. Empirical work from Ghana likewise reports that, although overall incidence may have declined following policy reforms, the poorest households remain disproportionately exposed to catastrophic out-of-pocket payments [30]. In a separate study from Kenya, financial constraints was a key barrier to accessing quality healthcare while 87% of respondents experienced substantial delays due to this constraint [31].

Moreover, findings show that as income rises, the likelihood of CHE at the 40% threshold declines sharply. This reflects the inverse equity phenomenon, where financial risk protection disproportionately benefits higher-income groups first, a pattern also documented in other LMICs where wealthier households are relatively insulated, while poorer households bear the brunt of out-of-pocket costs [32,33].

The treatment of BPH in Ghana imposes a significant economic burden, with more than 80% of respondents experiencing CHE. This highlights the financial vulnerability associated with treatment, particularly among low- and middle-income earners. The state is obliged to shield patients in these income brackets from catastrophic out-of-pocket payments. Comprehensive NHIS treatment coverage may play a vital role in ameliorating this burden. Nevertheless, awareness-raising programmes targeted at middle-aged men may prompt early treatment, which is likely to be associated with lower care costs. Another unique problem with urological services in Ghana is their centralization in urban areas, depriving peri-urban and rural communities of access. Consequently, these services are out of reach for patients in the peripheries, resulting in late presentation with its attendant complications and costs. Urological services should be decentralised by training more urologists and incentivising them to work in peri-urban and rural areas.

Our study relied on the minimum wage in the determination of productivity losses, this approach may have introduced bias since income variations occur across different sectors of the economy. Secondly, study estimates were based on data from a single institution; thus the generalizability of findings may be limited. This is even more the case because patients presenting to tertiary facilities like ours, tend to present with more severe disease with higher attendant cost. As a result, only few patients can afford specialist care. Our study population is therefore biased towards relatively wealthy patients with severe disease and may not be representative of the general population. Thirdly, our study evaluated cost from the patients' perspective; estimates from the societal or healthcare systems perspective are likely to be much higher. Another notable limitation is the absence of data on disease severity, as this may be a significant cost driver. Future studies should estimate disease severity as it could be an essential cost-determining factor. Nevertheless, these findings are worthwhile since they provide a baseline for future cost-of-illness studies on BPH in Ghana.

## Conclusion

The direct cost and productivity losses associated with BPH treatment had significant financial implications on patients and their households, with treatment modalities such as surgery and medications contributing to significant fractions of treatment cost. Despite the NHIS, patients still faced huge financial burden on direct medical cost with an average direct medical cost per patient being GHC 890.11 (USD 60.35). Transportation cost was the largest contributor to non-medical expenses. Productivity losses were substantial, as 30% of employed patients missed work, and 27.5% reported reduced working hours, leading to significant financial losses to patients and their households. In addition, this study found that CHE was disproportionately concentrated among the lowest-income groups, with all participants earning less than USD 136/month spending over 40% of their income on healthcare. Nearly all income categories exceeded the 10% threshold, underscoring widespread financial vulnerability among lower- and middle-income patients.

## Acknowledgments

We are grateful to the participants of this study who provided useful information to evaluate the cost implications of prostate hyperplasia management in Ghana.

## Author contributions

**Conceptualization:** Daniel Senanu Dadee-Seshie.

**Data curation:** Daniel Senanu Dadee-Seshie.

**Formal analysis:** Daniel Senanu Dadee-Seshie, Yakubu Alhassan.

**Investigation:** Daniel Senanu Dadee-Seshie.

**Methodology:** Daniel Senanu Dadee-Seshie, Yakubu Alhassan.

**Supervision:** Kekeli Kodjo Adanu.

**Validation:** Maxwell Ayindenaba Dalaba.

**Writing – original draft:** Daniel Senanu Dadee-Seshie.

**Writing – review & editing:** Sedem Godson Agbeteti, Banabas Kpankyaano, Aishah Fadila Adamu, Selase Kofi Adanu, Maxwell Ayindenaba Dalaba, Kekeli Kodjo Adanu.

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
