## [Decision Letter · Decision Letter 0]

15 Dec 2025

Dear Dr. Kekeli Kodjo Adanu,

Thank you for submitting your manuscript to PLOS ONE. After careful consideration, we feel that it has merit but does not fully meet PLOS ONE’s publication criteria as it currently stands. Therefore, we invite you to submit a revised version of the manuscript that addresses the points raised during the review process.

Please submit your revised manuscript in two weeks. If you will need more time than this to complete your revisions, please reply to this message or contact the journal office at plosone@plos.org . A letter that responds to each point raised by the academic editor and reviewer(s). You should upload this letter as a separate file labeled 'Response to Reviewers'.A marked-up copy of your manuscript that highlights changes made to the original version. You should upload this as a separate file labeled 'Revised Manuscript with Track Changes'.An unmarked version of your revised paper without tracked changes. You should upload this as a separate file labeled 'Manuscript'.

We look forward to receiving your revised manuscript.

Kind regards,

Xiuping Yu

Academic Editor

PLOS One

Reviewers' comments:

Reviewer's Responses to Questions

**Comments to the Author**

1. Is the manuscript technically sound, and do the data support the conclusions?

Reviewer #1: Yes

Reviewer #2: Yes

2. Has the statistical analysis been performed appropriately and rigorously?

Reviewer #1: Yes

Reviewer #2: Yes

3. Have the authors made all data underlying the findings in their manuscript fully available?

Reviewer #1: Yes

Reviewer #2: Yes

4. Is the manuscript presented in an intelligible fashion and written in standard English?

Reviewer #1: Yes

Reviewer #2: Yes

Reviewer #1: This study presents a cross-sectional cost-of-illness analysis of Benign Prostatic Hyperplasia (BPH) among 105 men attending the Urology Unit of Ho Teaching Hospital, Ghana. Using a bottom-up, patient-level costing approach, the authors estimate direct medical, direct non-medical, and productivity loss components from the patient perspective. The findings highlight that direct medical costs account for approximately 86% of total expenses, with non-medical and productivity losses contributing around 9% and 4%, respectively. The authors emphasize the growing economic burden of BPH and the need for policy attention in Ghana. This is a valuable and timely contribution to the limited literature on the economic burden of BPH in low- and middle-income countries. The study demonstrates methodological transparency and fills an important data gap for Ghana. The following comments could be addressed to improve the manuscript.

Comments:

The study addresses an important evidence gap—the economic burden of BPH in Ghana. However, data from a single tertiary hospital may not represent the broader population. Patients at referral centers often have more advanced disease or greater financial means to access specialist care. The discussion could better acknowledge how these selection factors may bias cost estimates and limit national generalizability.

The discussion references some studies from Europe and Asia but does not include countries of similar economic status to Ghana. Even brief inclusion of such studies would contextualize Ghana's results within a comparable economic and healthcare framework.

The manuscript’s main message is that BPH imposes a “significant economic burden,” but it lacks a clear linkage to policy or health system strategies (e.g., screening, early intervention, insurance coverage). Expanding this section to describe how the results could inform local reimbursement or urological service planning would strengthen its relevance.

Reviewer #2: Major Comments

1. BPH disease severity was not measured or adjusted, despite being a key driver of treatment choice and cost. This is a major limitation that should be emphasized.

2. Productivity loss estimation using minimum wage for all participants likely underestimates indirect costs and ignores income heterogeneity.

3. Catastrophic health expenditure (CHE) appears to be calculated using mean costs, rather than individual patient-level cost–income ratios, which may misclassify CHE. If feasible, re-estimate CHE using individual patient-level cost–income ratios.

4. Regression analysis is under-specified, excluding key clinical and treatment-related variables. Regression findings should be reframed as exploratory, and the limitations imposed by omitted variables should be explicitly discussed.

Minor Comments

1. Costs are reported in both GHS and USD using a single exchange rate. The authors should clarify whether any inflation adjustment was applied and state explicitly that USD conversions are for international comparison.

2. Standardize terminology for indirect costs and productivity losses throughout the manuscript.

3. Large standard deviations suggest skewed cost data; A brief justification for reporting means alongside medians would improve statistical clarity.

4. Strengthen policy implications: More concrete policy recommendations (e.g., NHIS medication coverage expansion, decentralization of urology services, transport subsidies) would strengthen the public health relevance.

**Do you want your identity to be public for this peer review?** For information about this choice, including consent withdrawal, please see our Privacy Policy

Reviewer #1: **Yes:**  Manisha Tripathi

Reviewer #2: No

---

## [Author Response · Author response to Decision Letter 1]

30 Dec 2025

The responses to comments have been added as an attachment.

---

## [Decision Letter · Decision Letter 1]

5 Jan 2026

Productivity losses and treatment cost of Benign Prostatic Hyperplasia in Ghana

PONE-D-25-56374R1

Dear Dr. Adanu,

We’re pleased to inform you that your manuscript has been judged scientifically suitable for publication and will be formally accepted for publication once it meets all outstanding technical requirements.

Kind regards,

Xiuping Yu

Academic Editor

PLOS One

Additional Editor Comments (optional):

Reviewers' comments:

Reviewer's Responses to Questions

**Comments to the Author**

Reviewer #1: All comments have been addressed

Reviewer #2: All comments have been addressed

2. Is the manuscript technically sound, and do the data support the conclusions?

Reviewer #1: Yes

Reviewer #2: Yes

3. Has the statistical analysis been performed appropriately and rigorously?

Reviewer #1: Yes

Reviewer #2: Yes

4. Have the authors made all data underlying the findings in their manuscript fully available?

Reviewer #1: Yes

Reviewer #2: Yes

5. Is the manuscript presented in an intelligible fashion and written in standard English?

Reviewer #1: Yes

Reviewer #2: Yes

Reviewer #1: (No Response)

Reviewer #2: (No Response)

**Do you want your identity to be public for this peer review?** For information about this choice, including consent withdrawal, please see our Privacy Policy

Reviewer #1: **Yes:**  Manisha Tripathi

Reviewer #2: No

---

## [Editor Report · Acceptance letter]

PONE-D-25-56374R1

PLOS One

Dear Dr. Adanu,

I'm pleased to inform you that your manuscript has been deemed suitable for publication in PLOS One. Congratulations! Your manuscript is now being handed over to our production team.

Kind regards,

on behalf of

Dr. Xiuping Yu

Academic Editor

PLOS One